

# A survey of Sybil attack countermeasures in IoT-based wireless sensor networks

Akashah Arshad, Zurina Mohd Hanapi, Shamala Subramaniam and Rohaya Latip

Faculty of Computer Science and Information Technology, Universiti Putra Malaysia, UPM Serdang, Selangor Darul Ehsan, Malaysia

## ABSTRACT

Wireless sensor networks (WSN) have been among the most prevalent wireless innovations over the years exciting new Internet of Things (IoT) applications. IoT based WSN integrated with Internet Protocol IP allows any physical objects with sensors to be connected ubiquitously and send real-time data to the server connected to the Internet gate. Security in WSN remains an ongoing research trend that falls under the IoT paradigm. A WSN node deployed in a hostile environment is likely to open security attacks such as Sybil attack due to its distributed architecture and network contention implemented in the routing protocol. In a Sybil attack, an adversary illegally advertises several false identities or a single identity that may occur at several locations called Sybil nodes. Therefore, in this paper, we give a survey of the most up-to-date assured methods to defend from the Sybil attack. The Sybil attack countermeasures includes encryption, trust, received signal indicator (RSSI), encryption and artificial intelligence. Specifically, we survey different methods, along with their advantages and disadvantages, to mitigate the Sybil attack. We discussed the lesson learned and the future avenues of study and open issues in WSN security analysis.

# INTRODUCTION

The Internet of Things (IoT) gained universal acceptance due to many applications for personal use and the community. IoT represents a collection of "Things" or embedded devices connected using various wireless technologies such as private and public networks (*Atzori, Iera & Morabito, 2010*). Based on the application domain, IoT applications are classifiable into six groups, for example, health care (*Zeb et al., 2016*; *Ambarkar & Shekokar, 2020*), environmental (*Kumari & Sahana, 2019*; *Behera et al., 2020*; *Zhuang et al., 2019*; *Jawad et al., 2017*), smart city (*Santos, Jimenez & Espinosa, 2019*; *Luo, 2019*), commercial (*Li & Cheffena, 2019*; *Khanna & Tomar, 2016*), IoT based robotic (*Roy Chowdhury, 2017*) and industry (*Zhu et al., 2020*).

Wireless sensor networks (WSNs) are essential subsets of IoT that have emerged as a core technology for a variety of data-centric applications. Almost all IoT network concepts are derived from WSNs. Both terms can be confusing at times, and there are many similarities and differences between IoT and WSN (*Pundir, Wazid & Singh, 2020*). IoT based WSN integrated with Internet Protocol (IP) allows any physical objects with sensors to be

Corresponding authors
Akashah Arshad,
gs47466@student.upm.edu.my
Zurina Mohd Hanapi,
zurinamh@upm.edu.my

connected ubiquitously and send real-time data to the server connected to the internet gateway. Sensor data is relayed to the base station and is saved in the cloud for future access (*Ala'Anzy & Othman, 2019*; *Sheron et al., 2020*). IoT-based WSN devices are powered by batteries that later can be replaced, which poses a significant challenge to application designers. To address these constraints in an IoT-based WSN, significant research has been conducted on managing network power consumption. Most existing research focus on extending the IoT network lifetime. The purpose of WSN is to gather data from the sensor node in a predetermined or random location and transmit the sensed data back to the base station.

The cumulative confirmed COVID-19 cases between 22 January and 12 October 2020 has reached 38,789,204 confirmed cases and has resulted in 1,095,097 deaths globally, as reported by WHO (2020). As a result, the need for monitoring systems is in great demand. The health of COVID-19 patients will be monitored continuously in an isolated room. Six per cent of them need to warded in the Intensive Care Unit (ICU) to save their lives, as reported by *El-Rashidy et al. (2020)*.

Hence, *Gupta et al. (2020)* foresee that smart sensors, actuators, devices, and data-driven applications can enable smart connected communities to strengthen the nations' health and economic postures to combat current COVID-19 and future pandemics efficiently. Flying drones regulated the quarantine and wearing of masks for public surveillance. Indoor isolation is made more accessible by robots and digital assistants. With the help of aware IoT devices, it is possible to track the origins of epidemics and ensure that patients follow important medical advice, as highlighted by *Fedele & Merenda (2020)*. However, from a security perspective, IoT networks are prone to sensor-based attacks based on a recent survey conducted by *Sikder et al. (2018)*. The authors also addressed IoT devices' vulnerability to sensor-based threats due to the lack of protection mechanisms to monitor the use of sensors by applications. An attack can be launched to the IoT based health application used to monitor COVID-19 patient. This security attack can put the patient's life in danger, where the attacker can manipulate the medical IoT devices.

Also, attackers can execute out local-scale attacks on individual critical devices that could include human life, such as the 2011 Stuxnet attack (*Kushner, 2013*), the late 2015 power-grid blackout of Ukraine (Dvorkin & Yury, 2020), the 2015 Jeep Cherokee attack (*Schneider, 2015*), the 2017 Brickerbot attack (*Radware, 2017*) and the 2018 Philips lightbulbs attack demonstration. In a world where every device is connected to IoT, these attacks have shown how catastrophic and diversified cybercrimes could be. Therefore, it is vital to detect Sybil attackers in WSN to prevent their malicious activities. In other words, Sybil attacks present a significant challenge for WSN, and improved defence mechanisms are required. We believe that the conducted survey work will help the researchers in Sybil countermeasures in WSN.

The recent survey covers the existing countermeasures to mitigate the WSN and IoT security attack (*Bhushan & Sahoo, 2017*). There is a literature review focus on Sybil attack countermeasures highlighted by *Vasudeva & Sood (2018)*, *Benkhelifa, Welsh & Hamouda (2018)* and *Gunturu (2015)* and its comparison shown in Table 1. A reader interested in Sybil countermeasures in an online network can read the following survey (*Al-Qurishi et*

Arshad et al. (2021), *PeerJ Comput. Sci.*, DOI 10.7717/peerj-cs.673

**Table 1   Existing literature review on Sybil Attack countermeasures.**

| Reference | Key management | Resource testing | Encryption | Trust | RSSI | Watchdog | TDOA | Artificial Intelligence | Incentive based | Application domain |
|---|---|---|---|---|---|---|---|---|---|---|
| This paper | ✓ | ✓ | ✓ | ✓ | ✓ | ✓ | ✓ | ✓ | X | WSN |
| *Vasudeva & Sood (2018)* | ✓ | ✓ | ✓ | ✓ | ✓ | ✓ | ✓ | X | X | WANET |
| *Bhise & Kamble (2016)* | ✓ | ✓ | X | X | ✓ | X | X | X | ✓ | Social network & WSN |
| *Gunturu (2015)* | ✓ | ✓ | X | ✓ | X | X | X | X | X | Social network |

**Table 2  Research questions.**

| RQ# | Research questions | Motivations |
|---|---|---|
| RQ1 | What is Sybil attack? | This question will help researchers understand the definition and the process of Sybil attack |
| RQ2 | Why is it important to focus on Sybil attack in IoT-based WSN environment | This question prove the purpose of countermeasure for Sybil attack |
| RQ3 | Where will new researchers concentrate on creating a new method? | This question is intended to help researchers look deeper into the research issue |
| RQ4 | How can the countermeasures in the Sybil attack attain better algorithms to thwart these attacks? | This question is intended to explain countermeasure use to thwart Sybil attack in obtaining optimal algorithms, identifying challenges and techniques |

*al., 2017*; *Alharbi et al., 2018*). However, there is no previous literature that reports any Sybil countermeasures based on artificial intelligence to the best of our knowledge. This paper provides a general review of up-to-date countermeasures used to mitigate the Sybil attack. Also, advantages, limitations and whether the existing proposed method is IoT ready are discussed.

The remainder of this paper is organised as follows. The "Survey Methodology" section illustrates the approach and methodology used in this literature review on the Sybil attack. In "Security Attack", we give a general overview of the Sybil attack. Next, we present the existing Sybil attack countermeasures in "Sybil attack countermeasures". In "Discussion", we discuss the comparisons of Sybil countermeasures in WSN and IoT. Finally, in the "Conclusions" section, we conclude the survey by summarising the paper and outlining future research directions.

# SURVEY METHODOLOGY

A systematic literature review (SLR) was carried out to examine countermeasures suggested by previous research studies to thwart Sybil's attack with the *Kitchenham et al. (2009)* benchmark, emphasising previous work related to countermeasures for attacks on Sybil. This research approach originated in the medical field to provide adequate knowledge for a repeatable study method (*Charband & Jafari Navimipour, 2016*; *Kupiainen, Mäntylä & Itkonen, 2015*). To guide the reader on why we need to focus on the Sybil attack, to discuss Sybil's critical principles and countermeasures as formalised in the following subsections, we chose four research questions.

## Research questions

The questions in this section were aimed at identifying the main issues and challenges along with countermeasures used for attacks by Sybil, including efficiency, end-to-end delay, overhead, packet delivery ratio, and detection metrics for Sybil attacks. This survey tries to address the following research questions (RQs), as shown in Table 2:

●RQ1: What is Sybil Attack?

●RQ2: Why is it important to focus on Sybil attack in IoT-based WSN environment? These two questions will prove the intent of countermeasure for Sybil Attack.

●RQ3: Where will new researchers concentrate on other methods of tackling the attack on Sybil? This problem aims to help the researcher focus on setting the direction of the proposed method.

●RQ4: How can the Sybil countermeasures achieve more robust algorithms to counteract those attacks?

This research question aims to explain how countermeasures are used to thwart Sybil's attack on achieving better algorithms, identifying challenges and techniques. The study needs are established after conducting a search query using suitable keywords, coming up with research questions, identifying the selection criteria, identifying the data retrieval, and conducting the quality evaluation. The aim of the survey could offer prepared answer and enlightenment for new researchers.

## Survey plan and organisation

The articles in this survey were acquired from most respected academic journals and also selected according to the checklist provided in (*Kitchenham et al., 2009*; *Vasudeva & Sood, 2018*) for the quality evaluation. The research articles acquired included IEEE, Elsevier, Springer, ACM, Wiley and MDPI, as these provided in-depth analysis. We started by filtering article by analysing the titles and abstracts. The entire research article is reviewed when the detailed information was not in the abstract. Hence, articles are selected in this analysis based on a detailed inquiry into the nature of their material and documents. This in-depth work enables us to have a consistent and thorough understanding of the countermeasures for Sybil in the IoT-based environment.

The paper analysis was undertaken between late January 2015 and July 2020 for the first filtering. Boolean functions (OR, AND) and specific keywords detailed by synonyms and alternative spellings were used to further investigate hundreds of papers in this area. (''Sybil'') and (''attack'' or ''attacks'')

Next, papers are filtered again to acquire papers more accurately related to the review context. The filtering process is to guarantee that no papers were overlooked in our review using the keywords search below:

(''Sybil'') and (''attack'' or ''attacks'') and (IoT OR ''internet of things'') and (WSN OR ''wireless sensor networks'') -''book'' -''conference''

## Eligibility criteria

Articles were evaluated based on the Quality Assessment Checklist (QAC) to be selected in our survey review list (*Kitchenham et al., 2009*; *Vasudeva & Sood, 2018*). The articles in this review that matched the research aim and objectives are selected according to the following criteria:

- Does the study paper identify the Sybil attack countermeasure methods?
- Is the methodology listed in the research paper?
- Do testing methodologies use the resources available for re-implementation (simulation or real system)?
- Does the research paper focus on WSN?
- Is the evaluation analysis done appropriately?

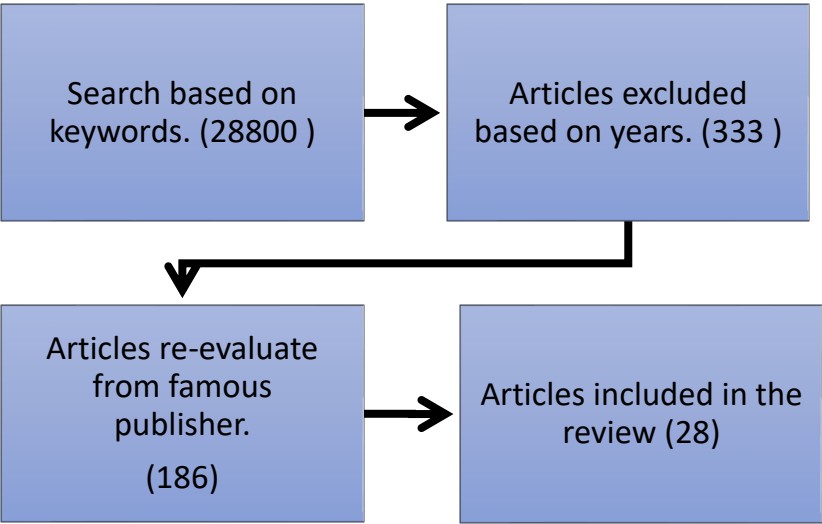

**Figure 1** **The selection of process of articles in the form of diagram.**

If "yes," the papers are chosen after the following conditions have been met:

- Any article that meets the criteria provided when there is a match in the keywords, the article is selected
- The article is filtered after going through the abstract and later will be recorded in the final list
- Articles related to the countermeasure of the attack on Sybil will be included.

## Data filtration and quality evaluation

The search engine for Google scholars was used to locate the primary studies with the automated search. The search led to the discovery of 28,800 articles that were considered significant for the study. Data for all publications cited, abstracts and keywords of all articles are further analysed in an Excel sheet resulted from phase by phase initial search. In this segment, we searched for keywords automatically and then find 372 journal articles and conference papers. Then, we included the year range 2010–2020, which is reduced to 333 journals. Then, we choose six famous publishers; 186 articles have been selected. Next, we checked whether any research papers satisfied the criteria or were ignored. When the abstract was found to be inadequate, the entire article was then checked, considering the requirements for inclusion or exclusion given above (*Kitchenham et al., 2009*). Then, according to the publication time, the number of 28 articles were selected and analysed. Figure 1 illustrates the procedure used to choose the articles for review. Researchers and scholars mostly publish their contributions in an established journal. Hence, conference papers have been excluded from this survey.

## Security attack

Generally, the security attack can be classified based on the attacker's objective, and on which layer the attack is carried out. The method of attack reviewed in the previous

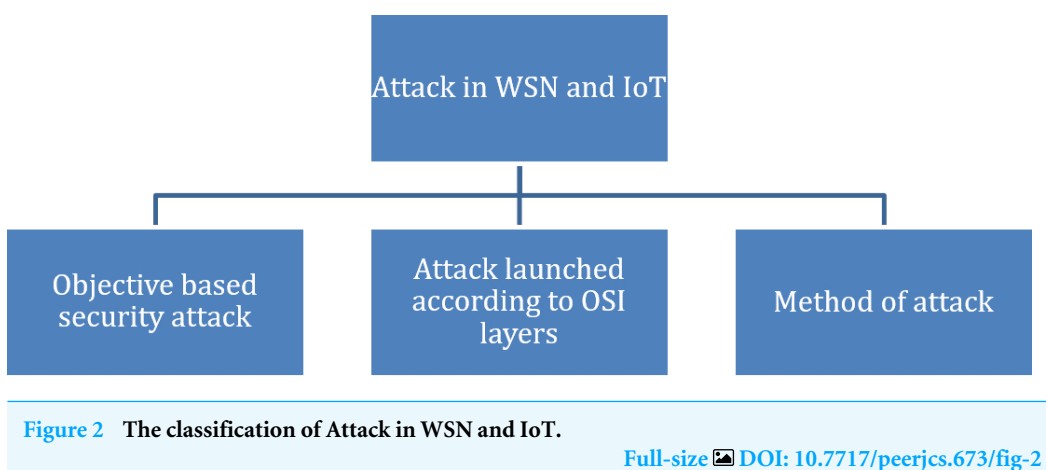

**Figure 2** The classification of Attack in WSN and IoT.

literature is shown in Fig. 2. Firstly, objective-based security attack can be divided into the passive and active attack. A passive attack can bring down the network, eavesdropping to collect personal information, node destruction, and node malfunction. In an active attack, the attacker has the objective to take down the targeted network to become useless. Several examples of active attacks can be further classified into flooding, jamming, Denial-of-Service (DoS), black hole, sinkhole, Sybil and wormhole. Secondly, the various passive and active attacks in WSN and IoT can be categorised according to OSI layers (*Butun, Osterberg & Song, 2020*; *Farjamnia, Gasimov & Kazimov, 2019*). Different types of attack in the IoT environment are described in (*Ahmad & Salah, 2017*). *Usman & Gutierrez (2018)* categorised the author's focus on wormhole attack and as well as other attacks are reviewed in (*Farjamnia, Gasimov & Kazimov, 2019*). Finally, the attack is categorised according to the attack method and how the malicious node can achieve its objective. Besides, the author also highlighted mitigation strategies against security attacks in Pervasive and Mobile Computing. Sybil, DoS, Hello and Sinkhole are layered network attacks in WSN that are still relevant in IoT environments (*Aufner, 2019*). Thus, it is applicable to any IoT devices which uses the communication layer to communicate.

Based on the earlier discussion on the attack, the countermeasures to security attack consist of prevention, detection, and mitigation. Firstly, the prevention method's main objective is to hinder the malicious attack from taking place in the first place. Secondly, detection countermeasures that are able to detect when there is a security breach in the network. The countermeasure method can identify the type of attack and launch the mitigation solution to reduce the damage done by malicious activity, as highlighted in Fig. 3. Hence, the mitigation method is the steps taken to reduce the after-effects of a security attack. Those three components are a complete protection framework and cannot be considered separately in the defence of WSNs and IoT against different types of attacks, as highlighted in (*Butun, Osterberg & Song, 2020*).

Such security attacks cause serious vulnerabilities to be routed inside the underlying network. Many attacks are less extreme, and others more severe (*Md Zin et al., 2015*). One of the first attacks in the WSN environment is the Sybil attack, leading to further security attack as a black hole and wormhole, as highlighted by *Murali & Jamalipour (2020)*. These

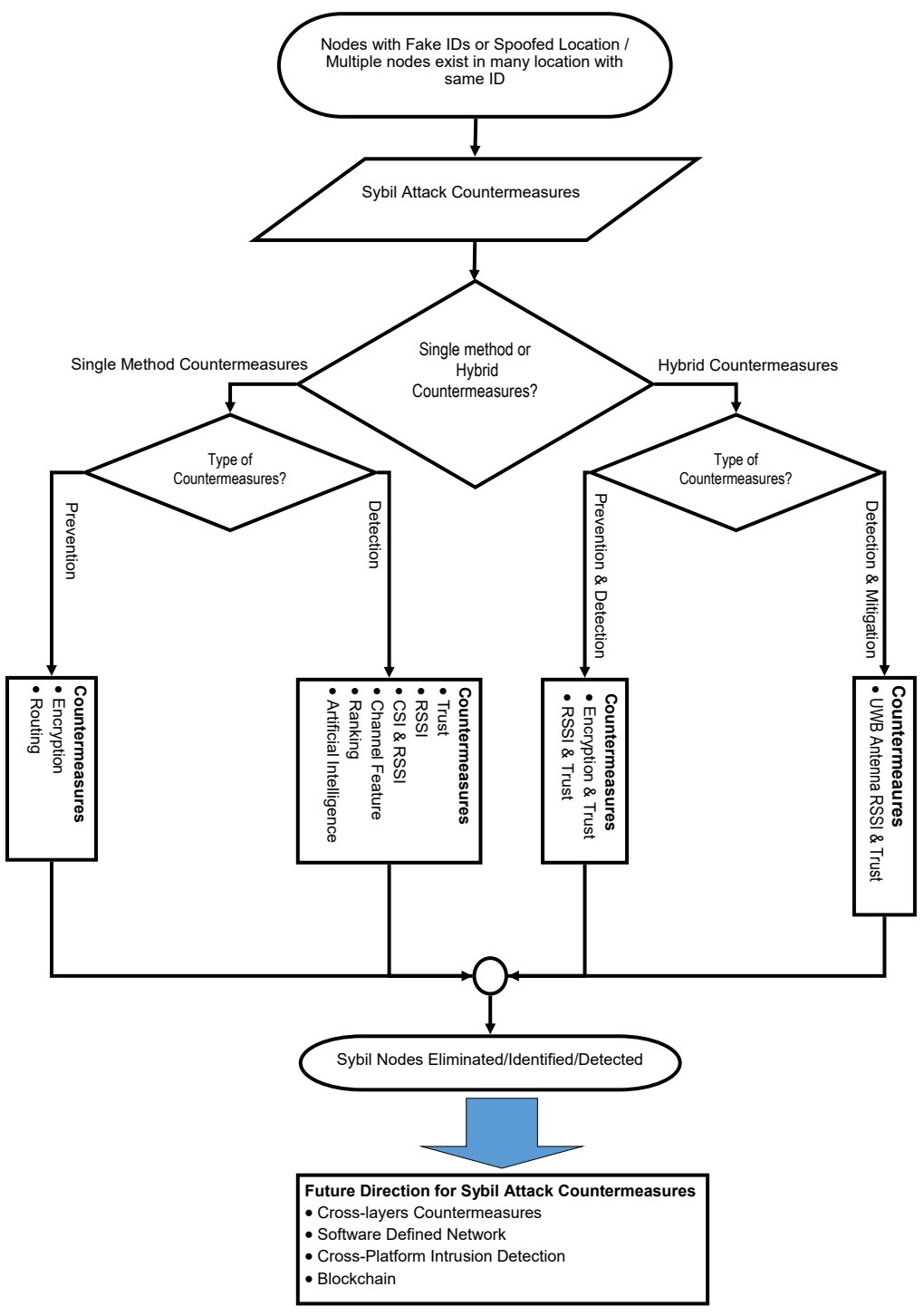

**Figure 3** **Sybil attack countermeasures framework.** .

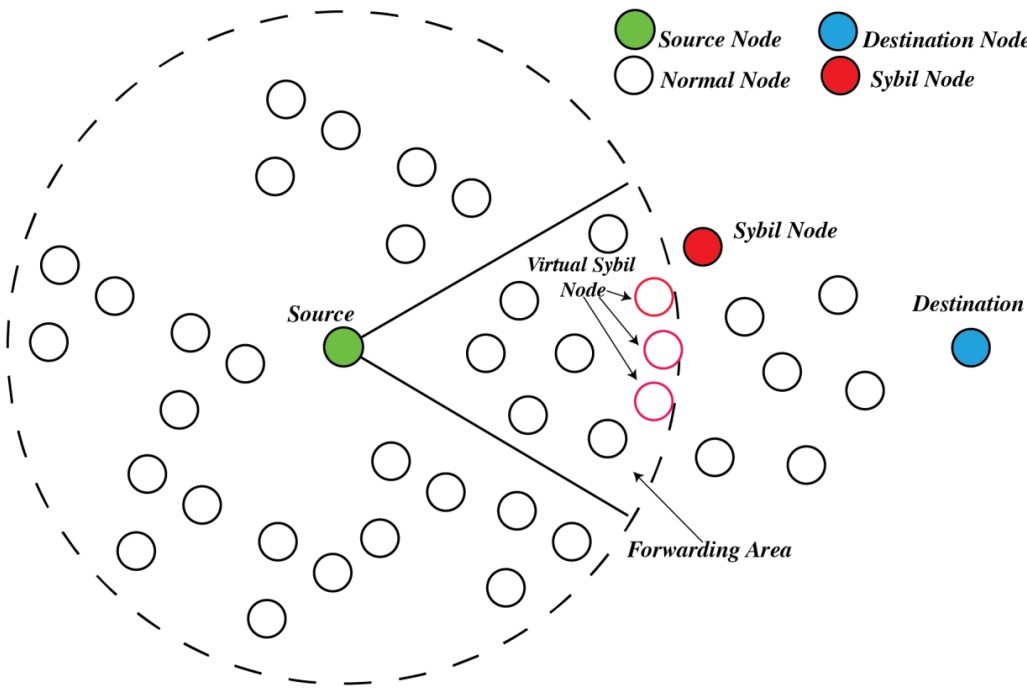

**Figure 4** The Sybil attack in geographic routing.

attacks can interrupt the operation of WSNs, which considers as IoT devices from collecting data sensors then stored in the cloud. Later, disruption caused IoT application like smart building, which houses many companies, to go haywire. Hence, this review paper focuses on the countermeasures for Sybil attack, which will be discussed later in the next section.

## Sybil attack

Sybil attack is defined by *Newsome et al. (2004)* when the malicious node can fake its own identity during an attack or steal the identity from working valid nodes. Sybil attack utilises fake identities to send false information, as highlighted by *Romdhani (2017)* and *Zhang et al. (2005)*. In an ad-hoc network, Sybil attack that utilises multiple fake identities are discussed in *Lv et al. (2008)*. In geographic routing, fake identities exist in the network with faked locations explored by *Sha, Gehlot & Greve (2013)* and *García-Otero et al. (2010)*, as shown in Fig. 4. Alternatively, a high-resource Sybil Attack can participate in the selection process by listening and transmitting its fake location during the protocol handshakes. *Newsome et al. (2004)* highlighted countermeasures for Sybil Attack, namely radio testing and random key pre-distribution. However, the author did not mention any limitation of the sensor network based on the listed method. *Goyal (2015)* and *John, Cherian & Kizhakkethottam (2015)* classified the Sybil Attack into a few countermeasures categories.

An attacker tries to get more attention from nearby nodes in this attack to intercept data packets. An attack that affects the process of packet delivery routed is called routing attacks. The simplest routing attack type is an altering attack in which the attacker modifies

the routing information by creating routing loops or fake error messages, as highlighted in *Mosenia & Jha (2017)*.

WSN security strategies can be broken down into two categories: prevention-based and detection-based. Due to restricted resources and a broadcast medium, mitigation methods such as encryption are challenging for WSNs. Also, all possible attacks will not be protected by the suggested cryptographic solutions. An attacker can easily obtain the symmetric key. The whole network is compromised since the attacker will decrypt all encrypted data using the symmetric key. The second line of defence is called the Intrusion Detection System (IDS), is important to detect malicious parties that try to use the weakness in the security and potential insecurities and detect attacks that have not been detected before (*Hidoussi et al., 2015*).

WSN nodes are deployed in the environment without any supervision. The unattended nature of WSNs, adversaries, can readily produce such Sybil attack. Severely compromised node and DOS attacks can interfere with the standard data delivery between sensor nodes and sink or even partition the topology, as *Shu, Krunz & Liu (2010)* highlighted. In the next chapter, we will address the Sybil Attack countermeasures suggested by the previous researchers.

## Sybil attack countermeasures

In this section, the countermeasures used to mitigate the Sybil attack can jeopardise humans' lives by monitoring IoT medical devices or other critical IoT applications. Radio resource testing (RTT) is a countermeasure that can distinguish direct forms of Sybil attack *Balachandran & Sanyal (2012)*. *Newsome et al. (2004)* stated that resource testing is a popular countermeasure to lower the probability of being attacked rather than eliminating Sybil attacks for good. *Ssu, Wang & Chang (2009)* proposed the RTT mechanism that assumed nodes in the network could not transmit and receive. Also, *Douceur (2002)* has proven that a trusted certification method can eliminate the Sybil Attack using a central authority. Some researchers proposed key management (*Paul, Sinha & Pal, 2013*; *Zhang et al., 2005*; *Eschenauer & Gligor, 2002*) or encryption for authentication using asymmetric key cryptography which is not suitable due to higher overhead and not scalable (*Boneh & Franklin, 2001*; *Zhu, Setia & Jajodia, 2006*; *Ssu, Wang & Chang, 2009*) Other methods include Sybil attacker detection by verifying neighbouring nodes' set, which caused higher communication overhead. Software-based attestation is a method where the verifier performs through various software or hardware challenges against its neighbouring node (*Steiner & Lupu, 2016*).

The radio signal is susceptible to interference and signal attenuation caused by the surrounding, which influences the precision of detecting a malicious node using RSSI-based and Time Difference of Arrival-based Scheme(TDOA) based countermeasures. *Chan & Ho (1994)* proposed two localisation methods, namely the estimation of TDOA and solving hyperbolic position. *Wen et al. (2008)* explained the TDOA ratio with the sender's identity. A Sybil node is detected once the beacon nodes calculated and find the same TDOA ratio for two different identities. These countermeasures are no longer in trend as many researchers are currently moving towards the proposed method, as shown

## Sybil Attack Countermeasures

**Figure 5** **Distribution of the Sybil attack countermeasures between 2010 and 2010.**

in Fig. 5. In most current literature, researchers focus on encryption and RSSI mechanisms that represent 29% of the solution provided for Sybil countermeasures. The rest of the solutions accounted for 14%, 14%, and 7% for trust, artificial intelligence and encryption hybrid. Lastly, 3% and 4% are accounted for rule-based anomaly and multi-kernel.

*Li & Cheffena (2019)* proposed a multi-kernel based expectation–maximization (MKEM) countermeasure for Sybil attacks. The innovative countermeasure analyses the radio resource of the sensor node to produce channel vectors. These channel vectors are comprised of the power gain and delay spread of the channel impulse response extracted from the received packet of the sensor node. In addition, a gap statistical analysis method was used to validate and EM method to summarise the detection results.

### Cryptography

Cryptography is a popular research area for WSN before IoT became the technological trend. Although cryptography requires lots of processing power, there is still an ongoing research area among researchers. *Kouicem, Bouabdallah & Lakhlef (2018)* highlighted two fundamental distribution approaches: key deterministic and probabilistic key distribution. In deterministic approaches, to provide maximum security coverage connection, each entity can make a secure link to others. However, the key management protocol becomes defenceless when under a security attack. Also, *Paul, Sinha & Pal (2013)* highlighted three types of cryptographic technique: creating and managing, distributing, and validating

keys for identities. Firstly, symmetric key distribution is a method using a similar key is utilised for encryption and decryption of the messages, namely Encryption Standard (AES), Rivest Cipher 4(RC4) and Triple Data Encryption Standard (3DES). However, key management problems and scalability issues are the main disadvantages of the symmetric key. WSN nodes are powered by batteries that are not suitable for implementing public-key cryptography due to high processing power and high network load when generating keys, distributing keys, and maintaining keys. Cryptographic devices are more likely to exposed to brute force attack. Asymmetric key distribution utilised the public key for encryption while the private key is utilised for decryption.

*Jain, Hussain & Kakarla (2020)* proposed a node authentication method for wireless sensor node to avoid security attacks and provide secure communication channels. The base station is responsible for generating the random value and a secret value to distribute among the sensor nodes. Each node is responsible for storing its secret and random value. *Zhang & Zhou (2010)* proposed using the Markle hash tree, trust values and message authentication codes for location verification algorithm. This approach works well with networks that are organised in a tree or hierarchical structure. This method falls in the encryption hybrid taxonomy shown in the previous pie chart diagram. *Claycomb & Shin (2011)* proposed a method based on a security policy that utilised key establishment to combine a group-based distribution model and identity-based encryption. *He, Zhang & Wei (2011)* proposed combining the merits of both public and symmetric cryptographic methods for key management in WSNs, where each node is configured with a public key system to establish end-to-end symmetric keys with other nodes like EDDK.

*Dong & Liu (2012)* proposed a scheme that deploys auxiliary nodes to execute the key establishment to help key establishment between sensor nodes. This method utili the secured Fuzzy Clustering Algorithm to determine the nodes that securely join the cluster. The cluster head oversees routing based on criteria, trust value and energy. *Kim & Kim (2013)* proposed a scalable and robust hierarchical key establishment scheme that enhances resilience against node capture, traffic analysis, and acknowledgement spoofing attack. In addition, this scheme provides periodic critical updates without communication costs for key transport. *Razaque & Rizvi (2017)* proposed a method to combat the Sybil attack, which comprises two novel algorithms. The first algorithm fragments the data to avoid detection from the malicious node. The second algorithm aims to provide authentication for nodes joining the network through encryption.

## Received signal strength indicator

Received Signal Strength Indicator (RSSI) was used to sense Sybil attack and accounted for about 26% in Fig. 5. RSSI remains the choice of researchers to mitigate against Sybil attack because it does not require special hardware to approximate the location of neighbours. There are many different approaches in RSSI to counter-attack and these vary between researcher. *Demirbas & Song (2006)* proposed a countermeasure method Sybil attack by only two receivers. To improve accuracy, *Wang et al. (2007)* came up with a countermeasure using RSSI from multiple neighbours instead of two neighbour nodes. Also, the status message can be used to validate the location in the hierarchical network

that utilises Jake Channel. *Zhong et al. (2004)* proposed the location verification based on RSSI signal using four or more detector nodes to detect the signals that can verify a node's location.

*Lv et al. (2008)* proposed a method for stationary WSNs called Cooperative Received Signal Strength (RSS) based Sybil Detection (CRSD) to estimate the distance between two identities and to locate the correlation of location between the unique identities of multiple neighbouring nodes. *Lazos & Poovendran (2005)* proposed a method that utilises the target node to determine its position using beacon information transmitted by both benevolent and malicious anchor nodes.

*García-Otero et al. (2010)* proposed innovative and lightweight location verification methods to detect and isolate Sybil attack. The distributed trust model is integrated with the routing protocol mainly to defend against routing attack. *Abbas et al. (2013)* utilise one neighbouring node to detect RSSI in mobile environments. Secure and Scalable Geographic, Opportunistic Routing with received signal strength (SGOR) is an opportunistic routing protocol proposed by *Lyu et al. (2015)*. This proposed trust method and a combination of calculating the difference of distance beacon messages and RSSI to detect the malicious nodes' fake location and defend against grey hole attack. The proposed method can defend against other attacks such as rushing, wormhole, replay, and collusion. However, this method's limitation is when the attacker has higher energy capacity and higher transmission power, which can easily deceive the sender about its location. *Kumari & Sahana (2019)* provided a framework using authentication and RSSI against Sybil attack. The RSSI values are calculated from the arrival angle, stored in the database at each node. The RSSI threshold value determines if the nodes fall within the safety zone and the precautionary zone. Also, the ant colony optimisation method was used to determine the optimised route for the packet to travel from source to destination. The second category assumes that a node can occur at one location at a specific time. *Raja & Maraline Beno (2017)* suggested another encryption approach using the Fujisaki Okamoto (FO) algorithm and their implementations. FO algorithm is an encryption method that offers good defence against Sybil attacks by using ID-based verification. In the proposed scheme, multiple performance metrics were analysed, especially the high energy consumption used as an indicator to sense Sybil attack in wireless sensor networks.

*Yuan et al. (2018)* presented a lightweight Approximate Point-in Triangulation Test (SF-APIT) algorithm that can pinpoint Sybil attacks in a wireless network in a distributed way using a range of free and iterative refinement-based methods. The individual node implementing the algorithm was based on RSS, which does not cost any overhead in WSN. Based on the node location, the node utilises three beacons in the triangulation method to calculate the possible combination overlapped triangle region, which can estimate the unknown node's location. Therefore, the centroid of the overlapping area is considered as the approximate location of this node. *Giri, Dutta & Neogy (2020)* proposed a countermeasure that protects the beacon node from a Sybil attack by implementing the information-theoretic approach. Any localisation algorithm can use this approach to provide protected localization in WSNs for the Sybil attack. Liu (2020) proposed an

improved RSSI-based Sybil attack detection scheme in WSNs. The proposed method able to quickly detect malicious nodes with minimal energy consumption.

The hierarchical topology of cluster network has many advantages in energy efficiency due to less communication, scalability, and routing. In addition, the proposed method utilised both RSSI and Channel State Information (CSI) to protect the hierarchical cluster network from Sybil attack. *Jamshidi et al. (2019)* proposed a lightweight method that consists of two algorithms for detecting Sybil node masquerading as cluster heads and cluster members. *Sarigiannidis, Karapistoli & Economides (2015)* proposed a secure communication mechanism for clustered WSNs based on the elliptic curve cryptography (ECC) that allows end-users to recover data collected confidentially. The proposed method has a firm reliance on historical records, making this approach not stable and durable. *Angappan, Sakthivel & Vishvaksenan (2020)* proposed a localized scheme for Sybil node detection called NoSad using RSSI value and intra-cluster communication, which can be deployed to the device. However, NoSad is not stable when there are a minimum of two Sybil node and cannot cater to mobility in WSN.

*Jan et al. (2015)* propose an innovative detection countermeasure for Sybil attack in a centralised clustering-based hierarchical network. Sybil nodes with fake identities are detected before the cluster to ensure that usage of the resources is optimised. The detection of Sybil nodes is achieved by analysing the received signal strength from any two high energy nodes. *Wang et al. (2018)* proposed a Sybil attack detection using CSI and a self-adaptive multiple signal classification algorithm RSSI for dynamic and static nodes in the clustered network.

## Trust

According to *Ishmanov & Bin Zikria (2017)*, there is not much research done on security attack detection based on unrelated criteria such as packet drop and packet modification. *Mawgoud, Taha & Khalifa (2020)* highlighted that trust could be set up automatically without personal interaction with previously unregistered and unknown peer neighbours in typical IoT scenarios.

*Karlof & Wagner (2003)* highlighted that the trust centre uses a key shared between two nodes for node verification to secure the network. *Zhan, Shi & Deng (2012)* proposed a trust management and encryption method that can detect and guess the future behaviour of a Sybil attacker. During next-hop selection, this trust information is vital to select a safe path to the destination.

*Zhan, Shi & Deng (2012)* proposed selecting the next hop based on trust and energy criteria. The energy watcher module calculates the energy cost for neighbouring nodes and the node's energy, where this information stored in the neighbourhood table. The energy watcher module also approximates the average energy required to route the packet from sender to destination. *Alsaedi et al. (2017)* proposed a method to detect Sybil attack based on name, location, and energy information for each time a new message was routed to the sender. The proposed method also uses a multi-level system where each rule to recognise a Sybil attacker is given to specific agents. These Sybil attackers engage in data aggregation at different stages to collude the aggregated data to disclose invalid data. Also, these malicious

nodes may modify and tamper with the timestamps of a message with multiple identities, which can cause havoc to synchronise local clocks in IoT devices. *Maddar, Kammoun & Youssef (2017)* proposed an innovative detection method for Sybil nodes with fake identities before cluster formation in a centralised clustering-based hierarchical network to optimize the usage of the resources. The detection countermeasure works by analysing neighbouring nodes for the received signal strength. *Jinhui et al. (2018)* proposed a method that can effectively predict energy consumption and increase the detection rate to detect malicious nodes.

*Liu, Abu-Ghazaleh & Kang (2007)* explained that landmarks are required to be trusted. All routing protocols are related to their mechanism of localisation and cannot be isolated from them. *García-Otero et al. (2010)* proposed a lightweight method that consists of localisation and intrusion identification techniques using a distrusted trust model to thwart several security attacks.*Prathusha Laxmi & Chilambuchelvan (2017)* proposed secure geographic routing (GSR), which has been modified from SecuTPGF. GSR's advantage is that it uses low computational power to combat security attacks such as spoofing and an assault on Sybil by introducing SHA-3 nodes and message authentication. *Zhou et al. (2015)* proposed a watchdog method that implements energy consumption optimisation while providing just enough security. The validation technique through a watchdog was able to defend against Sybil attack.

## Artificial intelligence

Intrusion detection systems are the example of artificial intelligence applications in the cybersecurity field. Cybersecurity solutions can distinguish between legitimate or malicious node through detailed traffic analysis. Cyberattacks were first detected with rule-based systems, which could detect attacks based on their signatures at the beginning of the Internet. Swarm Intelligence (SI) is a subdivision of artificial intelligence where the inspiration of this algorithm mimics biological swarms' intelligent behaviour in solving and simulating real problems. The SI algorithms are intended to investigate the concepts of simple individuals who can display sophisticated and complex swarm optimisation behaviours through collaboration, organisation, knowledge exchange, and learning between swarm members (*Kolias, Kambourakis & Maragoudakis, 2011*). These swarm intelligences can be categorized according to the year they were invented. Particle Swarm optimisation and Ant Colony optimisation was invented before the year 2000. Artificial Fish Swarm and Bacterial Foraging optimisation need further development to enhance, and Firefly optimisation and Artificial Bee Colony optimisation were widely used optimisation during the year 2000 until 2010. Pigeon inspired optimisation, Grey wolf optimiser, and Butterfly optimisation algorithm required further development.

*Prithi & Sumathi (2020)* proposed a method called Learning Dynamic Deterministic Finite Automata ($LD^2FA$) and PSO for intrusion detection, and the data is transmitted securely over-optimised paths. LD2FA-PSO got a 16% increase in throughput than cluster-based IDS, almost 70% rise in throughput than lightweight IDS, 6% and 32% increment in network lifetime over PSO and GLBCA, respectively; almost 30% and 54% improve in

network lifetime over GA and LDC, respectively. The energy consumed is almost 3% and 6% lower than PSO and GA, and 13% higher energy is consumed than LDC.

*Raghav, Thirugnansambandam & Anguraj (2020)* used swarm intelligence algorithms based on the bee to provide a secure routing scheme. The proposed routing mechanism utilise primary scout bee and secondary scout bee to carry out the secured and optimised routing. In many scenarios, it improves data efficiency while also providing security against flood, spoof, and Sybil attacks. Its disadvantages include that when the solution is close to the global optimum, it is possible to get stuck in the local optimum, resulting in stagnation.

## DISCUSSION

This paper has reviewed the countermeasures used to defend against a Sybil attack. Table 3 provides a comparative summary of the proposed method to countermeasure against a Sybil attack in term of its advantage, limitation, scalability readiness, classification of detection, and prevention. Besides security, scalability is also essential for deploying many devices under the IoT paradigm to become a major success (*Arellanes & Lau, 2020*). Security countermeasures should expand to many sensor nodes and intelligent devices (*Lu & Xu, 2019*). Comparing the proposed method will help future researcher evaluate and identify any research gap that will help them innovate or develop new countermeasures in the future. The proposed method to combat a Sybil attack is random key pre-distribution, cryptographic method, radio resource testing, RSSI localisation techniques, time difference of arrival (TDOA) localisation technique, neighbouring node information, trust, watchdog, RFID, clustering, and geographic routing.

Sybil attack countermeasures are of the simplified method due to the neighbouring node, and trust information is exchanges of control message between one or more nodes so that the sender can validate the identity of its neighbouring nodes. Also, this information is used as criteria's in selecting the best route from the sender to the destination nodes. Watchdog is used to monitor the neighbouring nodes in a centralised or decentralised scheme using the physical and data link layer. This information is used in selecting the best route for multi-hop routing.

Cryptographic and random key pre-distribution is implemented in the application layer, where its encryption and decryption process utilises the processing and memory resources. However, this authentication using asymmetric key cryptography has a higher overhead and not scalable. Also, the encryption process requires high computation and memory resources for the cryptography method and its attributes. The limitation for pre-distribution of the key is storing in databases that are vulnerable to attacks. However, the proposed method utilises high computational overhead, computational delays and a high load of control messages transmitted to nodes. Keys are store in databases that are vulnerable to attacks. One of the significant challenges in developing a lightweight key delivery network for sensor nodes with limited resources to support numerous protocols, applications and services at all IoT layers levels (*Gupta & Quamara, 2018*).

Radio resource testing, RSSI and TDOA measure the physical layers described by *Almas Shehni et al. (2017)* for the Sybil attack. RSSI and TDOA are two methods to locate Sybil

Arshad et al. (2021), *PeerJ Comput. Sci.*, DOI 10.7717/peerj-cs.673

**Table 3  Sybil attack countermeasures comparison.**

| Author, Year | Advantages | Disadvantages | Countermeasure Method | Type of Countermeasure | | | IoT Ready | Simulator |
|---|---|---|---|---|---|---|---|---|
| | | | | Prevention | Detection | Mitigation | | |
| *García-Otero et al. (2010)* | Energy efficient | High communication overhead | Trust | | √ | | | AWISSENET test-bed. |
| *Zhang & Zhou (2010)* | Low false positive rates at the same time even with high location inaccuracy | High communication overhead due to exchange of trust information | Encryption and Trust | √ | √ | | √ | Mathematical proof |
| *Claycomb & Shin (2011)* | A new approach to key establishment, which combines a group-based distribution model and identity-based cryptography | High complexity | Encryption | √ | | | | – |
| *He, Zhang & Wei (2011)* | EDDK utilizes encryption method is more advantageous in computation, communication, and storage | Weak resilience to node compromises. The shared pairwise key is static and is not secure against known-key attacks | Encryption | √ | | | | MATLAB |
| *Dong & Liu (2012)* | Extend the network lifetime | Does not support dynamic regular sensor node addition after initial deployment | Encryption | √ | | | | TelosB motes |
| *Zhan, Shi & Deng (2012)* | Energy efficient | Only suitable for static deployment | Trust | | √ | | √ | MATLAB |
| *Kim & Kim (2013)* | Better scalability and robustness | High computation cost | Encryption | √ | | | | Mathematical proof |
| *Lyu et al. (2015)* | Lightweight and distributed | A high probability of attack can occur at the sink and RSSI depends on the transmission of the node's energy | Trust & RSSI | √ | √ | | √ | OPNET |

Arshad et al. (2021), *PeerJ Comput. Sci.*, DOI 10.7717/peerj-cs.673

**Table 3** (*continued*)

| Author, Year | Advantages | Disadvantages | Countermeasure Method | Type of Countermeasure | | | IoT Ready | Simulator |
|---|---|---|---|---|---|---|---|---|
| | | | | **Prevention** | **Detection** | **Mitigation** | | |
| *Sarigiannidis, Karapistoli & Economides (2015)* | Lightweight and distributed | Mobility is not considered and not energy efficient | Using UWB antenna dan revoke malicious when detected | | √ | √ | | MATLAB |
| *Zhou et al. (2015)* | Lightweight and energy efficient | High complexity to the network | Watchdog using Trust | | √ | | | WSNET |
| *Jan et al. (2015)* | Improve network lifetime | The system fails if a malicious node able to imitate high energy node | RSSI | | √ | | √ | NA |
| *Saleem et al. (2016)* | Provide security through encryption with minimum processing time | Artificial immune system (AIS) - he major limitation of BIOSARP is that it requires time to develop the knowledge of the overall network during the initialization decryption - computational overhead | Encryption | √ | | | √ | NS2 |
| *Saleem et al. (2016)* | Lightweight and able to detect Sybil nodes accurately | Higher memory requirement | Energy trust calculation | | √ | | √ | OMNeT++ |

**Table 3** (*continued*)

| Author, Year | Advantages | Disadvantages | Countermeasure Method | Type of Countermeasure | | | IoT Ready | Simulator |
|---|---|---|---|---|---|---|---|---|
| | | | | **Prevention** | **Detection** | **Mitigation** | | |
| *Prathusha Laxmi & Chilambuchelvan (2017)* | Higher security due to the encryption method | High computation cost | Encryption | ✓ | | | ✓ | NetTopo |
| *Raja & Maraline Beno (2017)* | Higher security due to the encryption method | High computation cost | Encryption using Fujisaki Okamoto Algorithm | ✓ | | | | NS2 |
| *Maddar, Kammoun & Youssef (2017)* | Takes account attacks from different layers of OSI model | High communication overload | Trust calculation | | ✓ | | | MATLAB |
| *Razaque & Rizvi (2017)* | Higher security due to the encryption method | High communication cost | Encryption | ✓ | | | | NS3 |
| *Yuan et al. (2018)* | Lightweight | Not reliable due to radio interference and signal propagation | Localization with RSS signal | | ✓ | | | MATLAB |
| *Wang et al. (2018)* | Sybil attack detection system achieves high accuracy for both static and dynamic scenarios using CSI | It is not easily obtainable on the shelf NICs | CSI & RSSI | | ✓ | | | MATLAB |
| *Jamshidi et al. (2019)* | Lightweight | Not reliable due to radio interference and signal propagation | RSSI | ✓ | ✓ | | ✓ | J-SIM |
| *Li & Cheffena (2019)* | Higher accuracy and able to detect Sybil node | High processing load and communication overhead | Channel feature in terms of power gain and delay spread. | | ✓ | | | MATLAB |
| *Liu & Wu (2020)* | Rapid localization capability and high precision detection with low energy consumption. | Not reliable due to radio interference and signal propagation | RSSI | | ✓ | | ✓ | – |
| *Angappan, Sakthivel & Vishvaksenan (2020)* | Can apply to any resource-constrained WSN | It takes up memory if the device has limited | RSSI | | ✓ | | ✓ | NS2 |
| *Prithi & Sumathi (2020)* | Better throughput and network lifetime | PSO easily to fall into local optimum and low convergence rate in the iterative process. .. | Learning Dynamic Deterministic Finite Automata | | ✓ | | | MATLAB |

Arshad et al. (2021), *PeerJ Comput. Sci.*, DOI 10.7717/peerj-cs.673

| Author, Year | Advantages | Disadvantages | Countermeasure Method | Type of Countermeasure | | | IoT Ready | Simulator |
|---|---|---|---|---|---|---|---|---|
| | | | | Prevention | Detection | Mitigation | | |
| *Jain, Hussain & Kakarla (2020)* | Suitable for resource constraint nodes, there are no encryption keys involved in the security countermeasure | Hierarchical network topology suffers from non-uniform clustering, high energy dissipation, and less lifespan of the sensor nod | Authentication | √ | | | | AVISPA and Scyther tools |
| *Giri, Dutta & Neogy (2020)* | Successfully detect sybil attack and increase localization accuracy despite sybil attack. | High processing requirement | Localization with RSS signal | | √ | | | – |
| *Raghav, Thirugnansambandam & Anguraj (2020)* | Secure and optimised routing scheme with the help of bee algorithms. | High processing requirement and moderate scalability. It is easy to fall into the local optimum | Encryption & Bee Algorithm | √ | | | √ | MATLAB |
| *Bhushan & Sahoo (2020)* | Enhances energy efficiency and makes the network secure | Does not cover for availability | Trust and ACO | | √ | | | NA |

Attack by measuring signal strength and the distance between beacons. The RSSI method used less energy than other methods and did not require any special requirements or additional details. According to the studies, the distances between nodes from RSSI can be easily calculated based on RSSI information. RSSI-countermeasure methods are popular among researchers to detect Sybil Attacks (*Demirbas & Song, 2006*). However, the limitation of RSSI are susceptible to interference, environmental factors, the need for a beacon node, receiver system delay, non-line of sight transmission, and a malicious node with high power transmission could easily deceive the good node with its fake location and identity. The disadvantage of TDOA is that it is implemented in a highly dense network which can cause false detection of an honest node being detected as an attacker. An honest node's location is at the exact location as the detector node are the leading cause of false detection. Also, an attacker with a directional antenna could easily overcome being detected. These methods are not suitable for IoT devices that are mobile (*Wu & Ma, 2019*). RSSI has some limitation where there is no line of sight communication due to the obstruction of obstacle between a beacon node and a dumb node which caused the signal to get reflected from the surroundings (*Khan et al., 2013*). Hence, from the summary of countermeasures proposed by the previous researcher, future researchers should use RSSI due to its energy efficiency. To complement the limitation of RSSI, trust countermeasures based on energy due to energy heterogeneity of IoT devices should be combined with RSSI to enable the detection of Sybil malicious nodes.

Software attestation is a method where software routines are transmitted to the neighbouring node for validation. These routines are stored inside the memory, and neighbouring nodes are required to respond to the challenge within specific criteria like integrity validation in software and hardware, time duration, how software routines are read in the memory, and the interaction method. For example, the radio resource testing technique extracts the battery or energy level from these network devices. High energy devices are assumed to be malicious attacker nodes. However, this approach can cause the communication overhead to increase due to control packets for resource verification.

Figure 6 illustrates the proposed method that the researcher has developed from the year 2010 until 2020. The statistical charts show an increase in the encryption and trust method proposed by the researchers in 2017. In the year 2020, there is an increase in the proposed method using RSSI and less focus on encryption. Most of the artificial intelligence scheme proposed by the researchers in 2020 is used to optimise the routing process in complement to security. Hence, in the next section future researcher should try to integrate artificial intelligence to optimize the method such as cross layer, Software Defined Network (SDN), cross platform intrusion detection and blockchain.

## Lesson learned and future direction

WSN security is a hot research topic. There are many challenges and issues in WSN's security which future researcher can explore and provide a new solution. Specific requirements and constraints, such as low complexity and reliability, must be imposed on the provided solution. This section briefly discusses lessons learned from the previously proposed method and possible future directions for Sybil attack countermeasures.

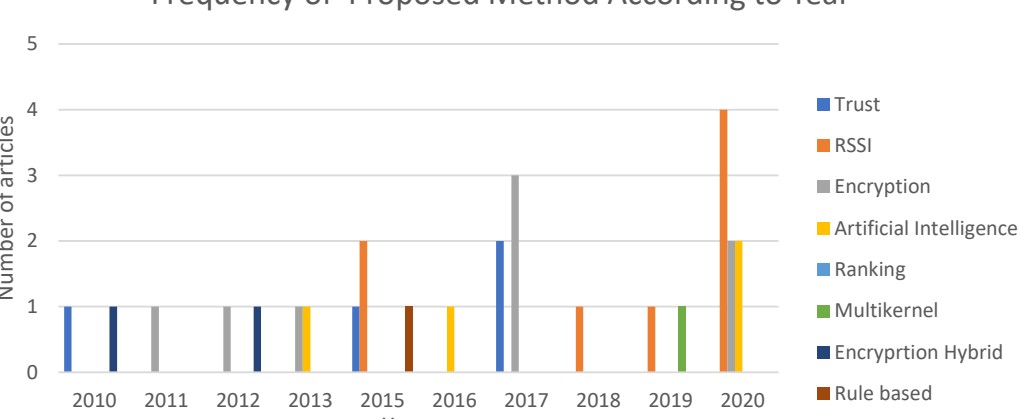

**Figure 6** Number of papers on Sybil countermeasures between 2010 and 2020.

## Cross-layer

Lesson learned: there is a possibility that an attack could be launched from the different layer during the communication process. Hence, this requires security countermeasures to handle cross-layer attacks and requires access to all information from multiple layers. Besides security, cross-layer information also beneficial in term of optimising energy efficiency

*Dhivya Devi & Vidya (2019)* discussed and explored the cross-layer design approaches that have been in WSN. For example, some proposed methods implement a cross-layer in detecting intrusion and routing (*Fatema & Brad, 2013*; *Umar et al., 2017*). The motivation to implement cross-layer design is due to the fact that it can optimise the network performance in the wireless sensor. The cross-layer design allows the ease of exchanging information between layers, which helps the WSN be energy efficient and increase QoS parameters. Based on the proposed method for Sybil attack, countermeasures surveyed, not many works of literature focus on security attack. The method of detecting a Sybil attack should incorporate the cross-layer approach to increase accuracy in detecting security attacks. The future researcher can utilise RSSI, which is lightweight from the physical layer with an upper layer such as Trust and Mobile agent for detecting a Sybil attack. A cross-layer method for detecting a Sybil attack with a mobile agent was proposed by *Gandhimathi & Murugaboopathi (2016)*. Cross-layer allows sharing of information among the MAC and network layers to optimise network performance. Also, this information can be utilised by a mobile agent to prevent a security attack. However, the proposed method to prevent a Sybil attack and another kind of attack increases the communication overhead.

## Software-defined network

Lessons learned: SDN and SDMN are the current trending research topics for 5G communication security. The exact security method for SDN and SDMN remain unexplored by researchers. With the deployment of SDN and SDMN in the communication, innovative techniques are needed in this area.

Apart from novel security solutions for IoT, there is a developing trend of SDN that allows reconfiguration of the network and central monitoring with possible centralised routing algorithm. This emerging paradigm opens up the researcher's door to develop a lightweight security framework running from the SDN controller, running at the central controller (*Hameed, FI & Hameed, 2019*).

## Cross-platform intrusion detection

Lesson learned: The IoT has gone from WSN where the sensor nodes are assumed to homogenous device with limited resources to heterogeneous devices with different capabilities but still limited in energy constraints. *Colom et al. (2018)* highlighted in the survey that the current trend of IDS is moving toward a universal and cross-platform method. The proposed method is able to handle device heterogeneity, scalability of IoT network and limitation.

Security and malware attack on the Internet could also be deployed in IoT due to various protocols utilised at every layer in the heterogeneous devices. The interoperability issues and lack of standard in IoT become a security challenge. Many IoT devices launched in the market have a security flaw as security was not the top priority and have not considered in the past. The previous IoT devices lack authentication methods or are able to detect or prevent an attack. A big challenge for intrusion detection methods to be deployed in the IoT environment due to the heterogeneity of devices. One example of a cross-platform intrusion detection; an innovative home application must retrieve information from personal healthcare sensing with a secure connection. Therefore, we need a quick, efficient and robust intrusion detection countermeasure to provide an undisruptive and continuous connection to multiple IoT platforms.

## Blockchain

Lesson learned: Blockchain is the latest decentralised distributed system technology designed and invented which included the Merkle trees for digital timestamps by *Bayer, Haber & Stornetta (1993)*. Proof of work, asymmetric cryptography, electronic signatures, and hash functions are all used in blockchain technology (*Lazrag et al., 2020*).

WSN sensor nodes are distributed and placed in an extreme and complex environment, so it is crucial to implement secure authentication between sensor nodes in WSNs (*Cui et al., 2020*). Blockchain is suitable for IoT with a hierarchical topology that has limited memory, computation, and energy. Merkle trees were incorporated into blockchain technology to provide efficient and reliable digital timestamps (*Dorri et al., 2017*). Blockchain has been applied in the security framework, and security countermeasure is still in the experimental phase, which will be the future direction of the research (*Mubarakali, 2021*)

# CONCLUSIONS

This paper discussed different countermeasures to defend the IoT-based WSN from the Sybil attack launched from various application domains. We have expanded on their modus operandi, advantages, and limitations of each countermeasure's categories. Although various researchers have proposed several countermeasures, there is no efficient

method to overcome most attacks with complete geographic routing accuracy. Also, we have observed that the trust mechanism is the most popular countermeasures for the Sybil attack from 2015 and 2020.

New research should investigate developing a framework that is lightweight to secure IoT network. Developing a secure framework for IoT, which consists of heterogeneous devices with different wireless technologies, is challenging. The development of a security framework for these IoT devices should consider IoT's scalability and resource constraint (*Razacheema, Alsmadi & Ikki, 2018*).

### Funding

This work was supported by Geran Putra Berimpak Universiti Putra Malaysia (9659400). The funders had no role in study design, data collection and analysis, decision to publish, or preparation of the manuscript.

### Grant Disclosures

The following grant information was disclosed by the authors:
Geran Putra Berimpak Universiti Putra Malaysia: 9659400.

### Competing Interests

The authors declare there are no competing interests.

### Author Contributions

- Akashah Arshad conceived and designed the experiments, performed the experiments, analyzed the data, performed the computation work, prepared figures and/or tables, authored or reviewed drafts of the paper, and approved the final draft.
- Zurina Mohd Hanapi, Shamala Subramaniam and Rohaya Latip conceived and designed the experiments, authored or reviewed drafts of the paper, and approved the final draft.

### Data Availability

This is a literature review; there is no raw data.

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
