# Peer review of "A survey of Sybil attack countermeasures in IoT-based wireless sensor networks"

_PeerJ Computer Science, doi:10.7717/peerj-cs.673_

## Round 0.1 · original submission · Major Revisions

In addition to the point-to-point explanation to the reviewer's comment; I DO ask the author to take a look at https://doi.org/10.1016/j.jnca.2018.07.006; https://link.springer.com/article/10.1007/s11276-019-01966-z; and other existing work on the survey of these attacks. The technical style of writing should be also improved professionally.

Reviewer 1 ·

Basic reporting

-

Experimental design

-

Validity of the findings

-

Additional comments

This is a survey article (or review article) addressing countermeasures in two types of attack, Sybil and Black Hole in IoT and WSN. While I appreciate the topic and scope, there are drawbacks in the rest of paper the authors need to improve.

1. The paper does not have a single focus. I could not know what the paper tries to focus either attack or countermeasure/ Sybil+Black hole or WSN/IoT. In additional, I do not know why the authors select to survey on these two attacks. Why two together, not only one? This can weaken the paper.
2. This is a survey paper, but the authors take too much effort on methodology and research questions while very few in the main knowledge (compared to the methodology). Nevertheless, the authors do not need to focus only articles published in five years. This could be indicated as a critical review article if the survey is more comprehensive with your own discussion. If the authors intend to propose a survey article, the authors need much more works on content gathering and statistics (quantitative analysis). Study from this survey paper https://ieeexplore.ieee.org/document/8792139

3. Overall, the body of knowledge (from the Security Attack section) is too limited where the reader almost learn nothing besides list of related literature with summary. For example, Sybil attack in Blockchain and electronic vote is not found. I suggest that the authors convert to a Review Article by focusing on one attack, studying on more literature (not only the last five years) especially on more recent articles, writing a critical discussion coming up with something the authors can use such as a framework, and removing the methodology section.

·

Basic reporting

No comment.

Experimental design

No comment.

Validity of the findings

No comment.

Additional comments

The paper provides an intensive survey on Sybil and Blackhole attacks and their countermeasures, in the context of IoT, with focus on relatively recent research work. It appears that they covered most of the major recent work in this area that I know of. The survey is comprehensive and did a good job in classifying the different approaches. The comparing between the major approaches is clear and well laid out.

---

## Round 0.2 · accepted · Accept

Thank you for submitting the paper to PeerJ CS and based on the decision provided by the reviewers, the paper now can be accepted.

Reviewer 1 ·

Basic reporting

-

Experimental design

-

Validity of the findings

-

Additional comments

The authors have made major improvement especially in terms of existing studies in a number of contexts, clear contribution, and performance comparison. This review paper is now much more comprehensive. Thus, it can be accepted.

* Some typos are found e.g. maskss in line 77. Language editing is needed.
* In the cover letter, the authors mentioned "Table 4", while this Table does not exist in the paper, and there is no need for Table 4.

·

Basic reporting

No comment.

Experimental design

No comment.

Validity of the findings

No comment.

Additional comments

The paper provides an intensive survey on Sybil and Blackhole attacks and their countermeasures, in the context of IoT, with focus on relatively recent research work. It appears that the authors covered most of the major recent work in this area that I know of. The survey is comprehensive and did a good job in classifying the different approaches. The comparison between the major approaches is clear and well laid out.

The revised version of the paper has improved the findings and quality.